# Indocyanine Green (ICG) in Robotic Gastrectomy: A Retrospective Review of Lymphadenectomy Outcomes for Gastric Cancer

**DOI:** 10.3390/cancers15204949

**Published:** 2023-10-11

**Authors:** Chul-Hyo Jeon, So-Jung Kim, Han-Hong Lee, Kyo-Young Song, Ho-Seok Seo

**Affiliations:** 1Division of Gastrointestinal Surgery, Department of Surgery, Uijeongbu St. Mary’s Hospital, College of Medicine, The Catholic University of Korea, Gyeonggi-do 11765, Republic of Korea; bfofsyr@naver.com; 2Division of Trauma and Surgical Critical Care, Department of Surgery, Uijeongbu St. Mary’s Hospital, College of Medicine, The Catholic University of Korea, Gyeonggi-do 11765, Republic of Korea; 3Division of Gastrointestinal Surgery, Department of Surgery, Seoul St. Mary’s Hospital, College of Medicine, The Catholic University of Korea, Seoul 06591, Republic of Korea; somang421@gmail.com (S.-J.K.); painkiller9@catholic.ac.kr (H.-H.L.); skygs@catholic.ac.kr (K.-Y.S.)

**Keywords:** stomach neoplasms, robotic surgical procedures, indocyanine green, gastrectomy, lymph node excision

## Abstract

**Simple Summary:**

Radical gastrectomy is pivotal for gastric cancer treatment with guidelines advocating for the dissection of at least 16 lymph nodes. However, the optimal number is debated, with some suggesting over 30 nodes. This research assessed the efficacy of ICG-guided robotic gastrectomy (an MIS technique) in ensuring thorough lymph node dissection. Analyzing data from 393 stage II or III gastric cancer patients, the study found that ICG-guided robotic surgery significantly increased the chances of achieving proper lymphadenectomy. This suggests its potential as a promising surgical approach for selected gastric cancer cases.

**Abstract:**

Radical gastrectomy is essential for gastric cancer treatment. While guidelines advise dissecting at least 16 lymph nodes, some research suggests over 30 nodes might be beneficial. This study assessed ICG-guided robotic gastrectomy’s effectiveness in thorough lymph node dissection. We analyzed data from 393 stage II or III gastric cancer patients treated at Seoul St. Mary’s Hospital from 2016–2022. Patients were categorized into conventional laparoscopy (G1, *n* = 288), ICG-guided laparoscopy (G2, *n* = 61), and ICG-guided robotic surgery (G3, *n* = 44). Among 391 patients, 308 (78.4%) achieved proper lymphadenectomy. The ICG-robotic group (G3) showed the highest success rate at 90.9%. ICG-guided robotic surgery was a significant predictor for achieving proper lymphadenectomy, with an odds ratio of 3.151. In conclusion, ICG-robotic gastrectomy improves lymphadenectomy outcomes in selected gastric cancer cases, indicating a promising surgical approach for the future.

## 1. Introduction

Globally, gastric cancer (GC) stands as the third most common cause of cancer-related mortality. In 2020 alone, an estimated 1 million new GC diagnoses led to 769,000 fatalities [1]. Notably, Korea has one of the world’s highest GC incidences, registering 41.4 to 51.9 cases per 100,000 individuals [2,3].

While the national cancer screening system’s introduction has notably elevated the detection rate of early gastric cancer, the significance of comprehensive lymph node (LN) dissection remains paramount [4,5,6]. The cornerstone of GC treatment is radical gastrectomy accompanied by lymphadenectomy, with postoperative adjuvant chemotherapy being an adjunctive option [7,8]. Current guidelines advocate for the examination of a minimum of 16 LNs. However, some research suggests the necessity for an even more extensive LN retrieval, highlighting an ongoing debate in the field [7,8,9,10,11].

The adoption of minimally invasive surgery (MIS) as a standard approach for gastric cancer is backed by a plethora of evidence [12,13]. Concurrently, the allure of robotic gastrectomy is on the rise, which is attributed to its advanced three-dimensional visualization, flexible instrument articulation, and precision [14]. Within the realm of MIS, various techniques are under exploration for optimal LN dissection. Notably, the integration of indocyanine green (ICG) with near-infrared (NIR) fluorescence imaging has demonstrated promising outcomes [15,16,17].

This study endeavors to ascertain the efficacy of robotic NIR imaging, combined with ICG, as a potential benchmark for ensuring thorough lymphadenectomy during radical gastrectomy for GC.

## 2. Materials and Methods

### 2.1. Patient Population and Data Collection

From January 2016 to February 2022, we enrolled 1948 patients who underwent curative radical gastrectomy for gastric cancer at Seoul St. Mary’s Hospital. Inclusion criteria encompassed pathologically confirmed primary gastric adenocarcinoma at stages II or III, R0 resection (absence of macroscopic or microscopic tumor remnants), laparoscopic or robotic surgical approaches, and complete data availability. We excluded patients who underwent open surgery, were at stages I or IV, or received preoperative chemotherapy or radiation therapy. After applying these criteria, 393 patients were included in the study. The enrollment process is depicted in Figure 1.

We collected patient demographic data and classified preoperative clinical characteristics using the Eastern Cooperative Oncology Group, and pathologic staging followed the eighth American Joint Committee on Cancer TNM guidelines [18].

Patients were stratified into three intervention groups:-Group 1 (G1): Laparoscopic surgery without the use of ICG-Group 2 (G2): ICG-guided laparoscopic surgery-Group 3 (G3): ICG-guided robotic surgery

### 2.2. Surgical Procedure, Endoscopic ICG Injection, and Intraoperative NIR Imaging

Specialized gastric cancer surgeons performed all surgeries, adhering to the Korean Gastric Cancer guidelines, utilizing either laparoscopic or robotic methods without open conversion [7]. 

Our fluorescent contrast agent was ICG (Dongindang Pharmaceutical), prepared as a sterile water solution at a concentration of 0.5 mg/mL. During surgery, we endoscopically injected 0.5 mL of this solution into the submucosal layer at four points around the primary tumor, totaling 2.0 mL (1.0 mg of ICG) (Figure 2A,B).

For the laparoscopic-ICG group, we captured NIR fluorescent images using the NOVADAQ fluorescence surgical system (Stryker Corp., Kalamazoo, MI, USA). This system seamlessly transitioned from visible light to NIR fluorescent imaging with a single click (Figure 3A). In the robotic-ICG group, surgeries were performed using the da Vinci Surgical System (Intuitive Surgical, Sunnyvale, CA, USA). Fluorescent images were integrated into the surgical view using infrared cameras attached to the robotic system (Firefly^®^; FLIR Systems, Wilsonville, OR, USA) (Figure 3B).

### 2.3. Determination of the Proper Lymphadenectomy (PL)

The number of examined lymph nodes (ELNs) correlates with lymph node metastasis; a higher ELN count typically indicates increased lymph node metastasis [6,19]. Consistently, the eighth edition TNM classification for GC advises dissecting a minimum of 16 lymph nodes [18]. Several studies have highlighted the positive relationship between ELN count and overall survival in GC patients. While the ideal ELN count remains a topic of debate, we established an ELN cut-off of 30 for stages II and III in our study. Cases exceeding 30 ELNs were deemed to have undergone PL.

Moreover, emerging evidence revealed the positive correlations between ELN count and overall survival of GC patients. By comparing ELN count to survival time, Okajima et al. proposed an optimal ELN count of ≥25; Gu et al. recommended an optimal ELN count of ≥16 for lymph node-negative GC and > 30 for lymph node-positive GC based on a stratified analysis of 7620 patients; and Zhao et al., in a multicenter study encompassing 4607 patients, reported that an ELN count of 30 or more is desirable [11,20,21,22]. The ideal cut-off for ELN count is debated in prior research, with some suggesting more than 40. However, emerging evidence has shown positive correlations between ELN count and overall GC patient survival. By comparing ELN count to survival, several studies proposed varying optimal counts. For stages II and III in this study, we set the ELN cut-off at 30. Cases with more than 30 ELNs were classified as having PL.

### 2.4. Statistical Analysis

We analyzed categorical variables using the chi-square or Fisher’s exact test, as appropriate. Continuous variables are presented as means ± standard deviations and were compared using Student’s *t*-test. A *p*-value < 0.05 was considered statistically significant. Both univariate and multivariate analyses employed the logistic regression model. All statistical evaluations were conducted using SPSS (ver.24; SPSS, Inc., Chicago, IL, USA) for Windows.

### 2.5. Ethical Approval and Consent to Participate

The institutional review board of the College of Medicine at the Catholic University of Korea approved this study (approval no. KC20RISI0593). We anonymized and de-identified all patient records before analysis.

## 3. Results

### 3.1. Patient Demographic and Clinicopathological Characteristics

Table 1 presents the clinicopathological characteristics of the patients. Out of the 391 patients in the study, 288 were categorized in group G1, 61 in G2, and 44 in G3. A higher proportion of patients aged 65 years or older with fewer comorbidities was observed in G3. Variables such as sex, performance status, preoperative body mass index, history of abdominal surgery, and extent of gastrectomy were distributed across the groups. The pathologic T stage was generally lower in G3 and higher in G1. Conversely, the pathologic N stage was lower in G1 and higher in G3. However, there was no statistical difference in the pathologic stage between the groups. The average number of ELNs was comparable across the groups (G1 vs. G2 vs. G3: 45.2 ± 20.3, 51.6 ± 24.5, 50.8 ± 19.6; *p* = 0.058).

### 3.2. Comparative Analysis of Patients Undergoing Proper Lymphadenectomy

In this study, proper lymphadenectomy (PL) was characterized as the retrieval of more than 30 ELNs. Out of the entire cohort, 308 patients (78.4%) achieved PL, while the remaining 85 (21.6%) did not meet this threshold. Impressively, the G3 group, which underwent ICG-guided robotic surgery, demonstrated a PL rate of 90.9%, thus surpassing the other groups. Subsequent analyses showed that patients with a higher N stage and more advanced cancer stages were more likely to achieve over 30 ELNs (Table 2). A closer assessment between the groups revealed a PL achievement rate of 76.0% for G1, compared to 80.3% for G2. The comparison between G2 and G3 yielded PL rates of 80.3% and 90.9%, respectively. While these rates differ, the difference was not statistically significant. However, when comparing G1 with G3, a significant difference in PL rates emerged, favoring G3 (*p* = 0.027), as detailed in Appendix A. 

### 3.3. Factors Influencing the Achievement of Proper Lymphadenectomy

Multivariate analyses, presented in Table 3, validated the ICG-guided robotic surgery as a significant determinant for exceeding 30 ELNs, with an odds ratio (OR) of 3.151 (95% CI: 1.074–9.245; *p* = 0.037). The N stage also emerged as a significant predictor for achieving proper lymphadenectomy. In contrast, variables such as age, sex, T stage, and resection extent did not show a significant impact (Table 3). Further scrutiny of our multivariate analysis, aimed at identifying specific factors associated with PL, revealed no significant differences between the G1 and G2 groups (*p* = 0.593) or between the G2 and G3 groups (*p* = 0.282). Interestingly, a comparison of G1 against G3 highlighted the G3 group as a significant predictor for achieving PL, demonstrating an odds ratio of 3.156 (CI 1.076–9.256, *p* = 0.036) (Appendix A).

## 4. Discussion

Our study delved into the efficacy of the ICG-NIR technique in MIS for gastric cancer patients. We aimed to discern the potential advantages of ICG-guided robotic lymphadenectomy. The insights we derived from our center’s robust data provide clarity on the ICG-NIR system’s role in robotic lymphadenectomy for advanced-stage gastric cancer patients [23,24].

The TNM staging system, despite the emergence of various innovative staging techniques, remains a cornerstone for assessing GC prognosis. Based on the count of positive lymph nodes, this system offers a reliable reflection of patient outcomes. The number of ELNs not only serves as a pivotal prognostic indicator for gastric cancer, but also impacts postoperative survival rates in other malignancies such as colorectal cancer [25,26,27]. Various factors can influence the count of ELNs, including the scope of the lymphadenectomy, the surgeon’s expertise, the depth of pathological examination of the lymph nodes, and individual surgical circumstances, such as the distribution of intra-abdominal fat and inherent variations in lymph node count between patients [11].

Building on previous research, our study emphasizes the pivotal role of the ELN count in shaping the prognosis of gastric cancer patients [10,11,20,22]. A diminished ELN count can precipitate ‘stage migration’, which might culminate in a more advanced diagnosis than initially projected [10]. The eighth edition of the AJCC gastric cancer staging system advocates for the examination of at least 16 lymph nodes [18]. However, there is a growing consensus, backed by studies, that a count of 30 or more nodes is optimal. For instance, Okajima et al. pinpointed an ideal ELN count of ≥25, while Gu et al. suggested a count of ≥16 for lymph node-negative GC and >30 for lymph node-positive GC, drawing from a stratified analysis of 7620 patients. Similarly, Zhao et al.’s multicenter study, which involved 4607 patients, posited that an ELN count of 30 or more is preferable [11,20,21,22].

The total number of histologically assessed nodes, irrespective of the positive nodal count, is a crucial prognostic indicator for gastric cancer, yet its practical application and interpretation remain subjects of debate [10]. In our study, we anchored our ELN cut-off on the foundational insights from prior research. Given that our threshold of 30 ELNs is on the higher side relative to earlier studies, we deemed it fitting to categorize prognostic groups based on this count in our analysis [20].

Intraoperative fluorescent lymphography, particularly with the use of ICG, has been recognized in prior studies as an essential tool for enhancing lymphadenectomy [16,28]. Recent advancements in ICG fluorescent imaging within laparoscopic and robotic surgeries suggest potential improvements in lymph node retrieval. Both laparoscopic and robotic gastrectomy methods have shown significant benefits from ICG fluorescence imaging. This imaging technique aids surgeons in the precise removal of potentially metastatic lymph nodes in gastric cancer, underscoring its value in selective fluorescence station-based lymphadenectomy [28,29]. ICG-aided fluorescent lymphography notably improves the precision of locoregional lymphadenectomy during robotic gastrectomy [28,30]. Moreover, using ICG fluorescent imaging minimizes the risk of lymph node tissue damage, prevents cancer cell spillage, reduces bleeding, and consequently enhances surgical outcomes and patient recovery [29,30,31,32].

In our study, we incorporated ICG-NIR techniques in MIS, encompassing both laparoscopic and robotic approaches. Our findings indicate that ICG-NIR application in robotic surgery considerably boosts the likelihood of achieving an ELN count of 30 or more, representing PL. This is in contrast to the results from the laparoscopy without the ICG group or the ICG-guided laparoscopy group. While there were no significant statistical variances among the groups, an ELN count of at least 45 consistently emerged, marking a standard for surgical quality at our institution. Importantly, we observed neither complications from ICG endoscopic administration, nor any intraoperative or postoperative issues related to NIR imaging.

This study, a retrospective analysis, aimed to evaluate the efficacy of ICG fluorescent imaging in lymphadenectomy during robotic gastric surgery versus laparoscopy. Our comprehensive assessment revealed that ICG significantly bolsters lymphatic mapping in robotic gastric cancer surgeries. Building on this, insights derived from earlier work by Prof. Song, who led the robotic surgeries in our investigation, underscore the benefits of robotic surgery. Notably, the robotic approach offers a three-dimensional visual perspective, minimizes hand tremors, and provides superior maneuverability in the abdominal space. Such advantages, especially the robotic arm’s flexibility, have been linked to a more successful retrieval of lymph nodes, particularly in the challenging suprapancreatic region. Comparatively, while laparoscopic instruments might be constrained by the complex contours of the abdomen, the robotic platform’s agility ensures precise access and movement. In subsequent evaluations, robotic gastrectomy yielded higher lymph node retrievals and reduced blood loss, albeit with extended operation times and increased costs. The clarity and precision offered by the robotic system, alongside its ability to protect blood vessels, play a pivotal role in these outcomes [23,33,34,35].

The ICG-guided robotic surgery group exhibited a markedly higher lymphadenectomy success rate compared to the control group. Furthermore, ICG stands out as a cost-effective alternative, especially when compared to carbon nanoparticles. Its safety record remains commendable with no reported adverse events, making it a viable choice for lymphadenectomy due to its affordability, user-friendliness, and safety [29]. Diving deeper into the factors influencing proper lymphadenectomy, our extended analysis did not reveal statistically significant differences between groups G1 and G2, or G2 and G3. However, a notable difference was observed when comparing G3 to G1. This suggests that while the individual impact of ICG imaging or robotic surgery may not be readily discernible, their combined utilization in the ICG-guided robotic surgery group significantly elevates the success rate of proper lymphadenectomy. Specifically, when juxtaposed against laparoscopic surgery without the use of the ICG group, the synergy of these methods becomes evident. This lends further credence to our stance that ICG-robotic gastrectomy can potentially enhance lymphadenectomy outcomes in selected gastric cancer cases, marking it as a promising surgical approach for forthcoming applications.

However, our study does have limitations. Being a single-institution, retrospective analysis focused on Asian participants, the findings might not be universally applicable. Gastric cancer patients in Western regions often differ in characteristics such as obesity rates, disease stages, and treatment methods. Thus, extrapolating these results to a global context requires prudence. Another challenge is the ongoing debate over the ideal ELN cut-off. To address this, we referenced literature reviews from other publications to determine a suitable ELN threshold for gastric cancer. Despite its retrospective nature, our study underscores the potential of ICG in enhancing lymphadenectomy accuracy during robotic gastrectomy.

## 5. Conclusions

In conclusion, ICG-guided robotic gastrectomy appears to be more effective in achieving more than 30 ELNs and performing PL than conventional laparoscopic gastrectomy in selected cases. ICG guidance, thus, presents a promising avenue in robotic gastric cancer surgeries, enabling surgeons to achieve higher lymphadenectomy success rates without compromising surgical safety.

## Figures and Tables

**Figure 1 cancers-15-04949-f001:**
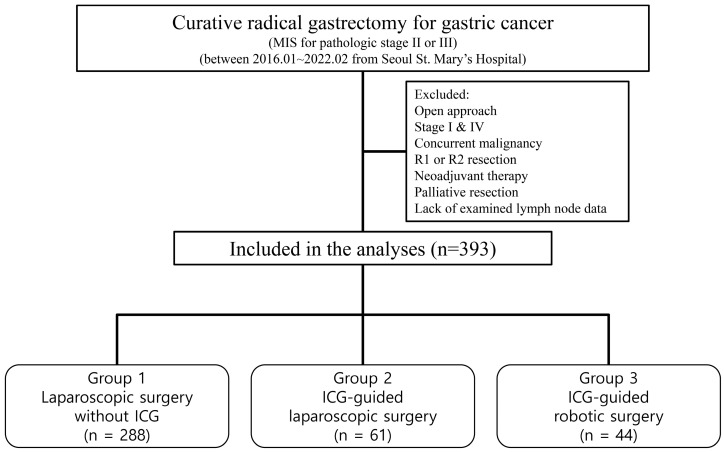
Flow chart illustrating patient selection.

**Figure 2 cancers-15-04949-f002:**
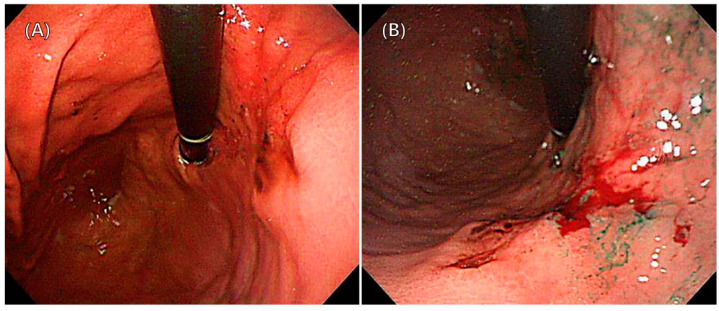
Endoscopic peritumoral ICG injection. (**A**) Before the injection. (**B**) After the injection.

**Figure 3 cancers-15-04949-f003:**
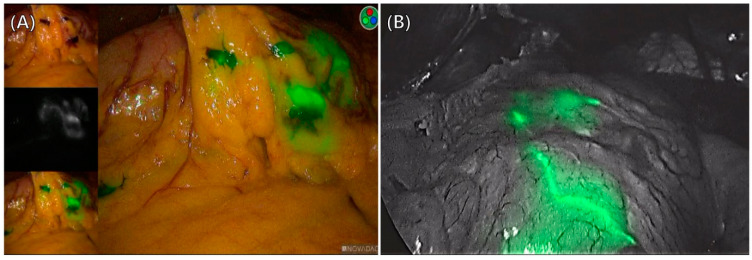
Visualization of lymph nodes using ICG during surgery. (**A**) Laparoscopic gastrectomy. (**B**) Robotic gastrectomy.

**Table 1 cancers-15-04949-t001:** Demographic and clinicopathological profiles.

Variables, *n* (%)	G1 (*n* = 288)	G2 (*n* = 61)	G3 (*n* = 44)	*p*-Value
Age (years)				0.001
<65	137 (47.6)	31 (50.8)	34 (77.3)	
≥65	151 (52.4)	30 (49.2)	10 (22.7)	
Sex				0.439
Male	193 (67.0)	46 (75.4)	30 (68.2)	
Female	95 (33.0)	15 (24.6)	14 (31.8)	
ECOG				0.128
0–1	272 (94.4)	60 (98.4)	44 (100.0)	
≥2	16 (5.6)	1 (1.6)		
Preoperative BMI (kg/m^2^)				0.141
<23	121 (42.3)	20 (32.8)	13 (29.5)	
≥23	165 (57.7)	41 (67.2)	31 (70.5)	
Comorbidity				0.001
Present	207 (71.9)	25 (41.0)	24 (54.5)	
Absent	81 (28.1)	36 (59.0)	20 (45.5)	
History of abdominal surgery				0.671
Present	72 (25.0)	12 (19.7)	11 (25.0)	
Absent	216 (75.0)	49 (80.3)	33 (75.0)	
Extent of gastrectomy				0.123
STG	231 (80.2)	43 (70.5)	31 (70.5)	
TG	57 (19.8)	18 (29.5)	13 (29.5)	
pT stage				0.039
T1	38 (13.2)	3 (4.9)	9 (20.5)	
T2	49 (17.0)	14 (23.0)	10 (22.7)	
T3	133 (46.2)	34 (55.7)	22 (50.0)	
T4	68 (23.6)	10 (16.4)	3 (6.8)	
pN stage				0.039
N0	70 (24.3)	13 (21.3)	9 (20.5)	
N1	67 (23.3)	26 (42.6)	9 (20.5)	
N2	87 (30.2)	16 (26.2)	14 (31.8)	
N3	64 (22.2)	6 (9.8)	12 (27.3)	
pTNM stage				0.277
Stage II	175 (60.8)	42 (68.9)	31 (70.5)	
Stage III	113 (39.2)	19 (31.1)	13 (29.5)	
ELN Count	45.2 ± 20.3	51.6 ± 24.5	50.8 ± 19.6	0.058
≥16	285 (99.0)	61 (100.0)	43 (97.7)	0.518
≥30	219 (76.0)	49 (80.3)	40 (90.9)	0.049

ECOG—Eastern Cooperative Oncology Group performance status; BMI—body mass index; STG—subtotal gastrectomy; TG—total gastrectomy; ELN—examined lymph node.

**Table 2 cancers-15-04949-t002:** Patient characteristics influencing proper lymphadenectomy.

Variables, *n* (%)	PL (-) (*n* = 85)	PL (+) (*n* = 308)	*p*-Value
ICG Injection			0.063
Present	16 (18.8)	89 (28.9)	
Absent	69 (81.2)	219 (71.1)	
Intervention group			0.049
G1	69 (24.0)	219 (76.0)	
G2	12 (19.7)	49 (80.3)	
G3	4 (9.1)	40 (90.9)	
pT stage			0.273
T1	16 (18.8)	34 (11.0)	
T2	16 (18.8)	57 (18.5)	
T3	38 (44.7)	151 (49.0)	
T4	15 (17.6)	66 (21.4)	
pN stage			0.001
N0	27 (31.8)	65 (21.1)	
N1	17 (20.0)	85 (27.6)	
N2	34 (40.0)	83 (26.9)	
N3	7 (8.2)	75 (24.4)	
pTNM stage			0.009
Stage II	64 (75.3)	184 (59.7)	
Stage III	21 (24.7)	124 (40.3)	

PL—proper lymphadenectomy; ICG—indocyanine green.

**Table 3 cancers-15-04949-t003:** Univariate and multivariate analysis of factors contributing to proper lymphadenectomy.

Variables	Univariate			Multivariate		
	OR	95% CI	*p*-Value	OR	95% CI	*p*-Value
Older age (vs. <65)	0.754	0.466~1.221	0.251			
Male sex (vs. Female)	0.988	0.589~1.655	0.962			
ECOG 2–4 (vs. 0–1)	1.542	0.528~4.504	0.429			
Higher BMI (vs. <23)	1.060	0.648~1.734	0.818			
No comorbidity (vs. present)	0.599	0.347~1.034	0.066			
No history of abdominal surgery (vs. present)	0.761	0.443~1.309	0.324			
TG (vs. STG)	0.624	0.332~1.171	0.142			
T stage						
T1	Ref			Ref		
T2	1.676	0.744~3.779	0.213	1.175	0.461~2.991	0.736
T3	1.870	0.936~3.738	0.076	2.284	0.957~5.446	0.063
T4	2.071	0.915~4.687	0.081	2.189	0.890~5.388	0.088
N stage						
N0	Ref			Ref		
N1	2.077	1.044~4.130	0.037	2.150	1.061~4.356	0.034
N2	1.014	0.556~1.849	0.964	0.995	0.543~1.824	0.987
N3	4.451	1.82~10.894	0.001	4.414	1.794~10.862	0.001
Intervention group						
G1	Ref			Ref		
G2	1.287	0.647~2.557	0.472	1.285	0.634~2.606	0.487
G3	3.117	1.061~9.153	0.034	3.151	1.074~9.245	0.037

OR—odds ratio; CI—confidence interval; ECOG—Eastern Cooperative Oncology Group performance status; BMI—body mass index; TG—total gastrectomy; STG—subtotal gastrectomy.

## Data Availability

Data is unavailable due to privacy and ethical restrictions.

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
