# Peer review of "Indocyanine Green (ICG) in Robotic Gastrectomy: A Retrospective Review of Lymphadenectomy Outcomes for Gastric Cancer"

_cancers, 2023, doi:10.3390/cancers15204949_

Round 1
Reviewer 1 Report
In this paper Chul Hyo Jeon et al. aims to asses the efficacy of ICG-guided robotic gastrectomy, compare to laparoscopic gastrectomy and ICG-guided laparoscopic gastrectomy, in ensuring adeguate lymph node dissection. The study is well conducted with a good number of patients. The results are well presented and consistent. I have only one question: in which phase of the gastrectomy and how in your opinion the robotic platform help the most in the lymphadenectomy?
Author Response
Reviewer 1
In this paper Chul Hyo Jeon et al. aims to asses the efficacy of ICG-guided robotic gastrectomy, compare to laparoscopic gastrectomy and ICG-guided laparoscopic gastrectomy, inensuring adeguate lymph node dissection. The study is well conducted with a good number of patients. The results are well presented and consistent. I have only one question: in which
phase of the gastrectomy and how in your opinion the robotic platform help the most in the lymphadenectomy?
Response: Thank you for your insightful comments and pertinent question regarding our work. In a prior study conducted by the author Song, who spearheaded the robotic surgeries in our current study, several distinct advantages of robotic surgery were highlighted. Chief among them was the provision of a three-dimensional field of view, mitigation of the operator's hand tremors, and enhanced intra-abdominal maneuverability due to the flexible wrist-like motion of the robotic instruments. Interestingly, while there were no notable differences in gastric resection or reconstruction between robotic and laparoscopic surgery groups in that study, the robotic group demonstrated a superior count of retrieved lymph nodes, especially from the suprapancreatic area. This advantage is attributed to the dexterity and multi-joint capabilities of the robotic arm, allowing it unparalleled access and movement within the complex contours of the abdominal cavity. For example, the concave shape of the upper part of the pancreas poses challenges for straight laparoscopic instruments, but the flexibility of the robotic arm offers enhanced precision in such areas. [J Gastric Cancer. 2021 Sep;21(3):308-318], [Yonago Acta Med. 2020 May 18;63(2):99-106]
In another related study, robotic gastrectomy (RG) was associated with a significantly higher number of retrieved lymph nodes, particularly from the supra-pancreatic region, when compared to laparoscopic gastrectomy (LG). This was alongside other benefits like reduced blood loss. However, RG also presented with longer operation durations and elevated hospitalization costs. It's hypothesized that the tremor filtration capability and superior 3D visual field offered by the robotic platform, coupled with the ability to provide precise anatomical positioning without damaging blood vessels, account for these differences. [Surg Today. 2020 Sep;50(9):955-965], [Surg Endosc. 2022 Jan;36(1):185-195]
Given the results from these studies and the surgical outcomes observed in our current work, we concur that the robotic platform offers distinct advantages, especially in the dissection of supra-pancreatic lymph nodes during gastrectomy.
We have added the following section to the discussion (page 9.Line 216-227).
[Building on this, insights derived from earlier work by Prof. Song, who led the robotic surgeries in our investigation, underscore the benefits of robotic surgery. Notably, the ro-botic approach offers a three-dimensional visual perspective, minimizes hand tremors, and provides superior maneuverability in the abdominal space. Such advantages, espe-cially the robotic arm's flexibility, have been linked to more successful retrieval of lymph nodes, particularly in the challenging suprapancreatic region. Comparatively, while lap-aroscopic instruments might be constrained by the complex contours of the abdomen, the robotic platform's agility ensures precise access and movement. In subsequent evaluations, robotic gastrectomy yielded higher lymph node retrievals and reduced blood loss, albeit with extended operation times and increased costs. The clarity and precision offered by the robotic system, alongside its ability to protect blood vessels, play a pivotal role in these outcomes]
Reviewer 2 Report
Authors investigated the efficacy of robotic NIR imaging, combined with ICG, as a potential benchmark for ensuring thorough lymphadenectomy during radical gastrectomy for GC(page.2,line.55-57), and concluded that ICG-robotic gastrectomy improves lymphadenectomy outcomes in selected gastric cancer cases, indicating a promising surgical approach for the future.(page.1,line.30-32).
However, there were some questions to reach the conclusion. It was unclear which was the significant factor of proper lymph node dissection , ICG imaging or robotic method, or both?
Therefore, authors were recommended to make analysis more in detail as below .
Authors didn't make analysis of the proper lymph node dissection between G1 and G2. And, authors didin't also make analysis between G2 and G3.
Authors were recomended to make analysis as above respectively , and then, to estimate which had the significant efficacy , ICG imaging or robotic method and to discuss about them to reach conclusions.
Author Response
Reviewer 2
Authors investigated the efficacy of robotic NIR imaging, combined with ICG, as a potential benchmark for ensuring thorough lymphadenectomy during radical gastrectomy for GC (page.2,line.55-57), and concluded that ICG-robotic gastrectomy improves lymphadenectomy outcomes in selected gastric cancer cases, indicating a promising surgical approach for the future. (page.1,line.30-32).
However, there were some questions to reach the conclusion. It was unclear which was the significant factor of proper lymph
node dissection , ICG imaging or robotic method, or both?
Therefore, authors were recommended to make analysis more in detail as below .
Authors didn't make analysis of the proper lymph node dissection between G1 and G2. And, authors didin't also make analysis
between G2 and G3.
Authors were recomended to make analysis as above respectively , and then, to estimate which had the significant
efficacy , ICG imaging or robotic method and to discuss about them to reach conclusions.
Response: We're grateful for the constructive feedback provided by Reviewer. In response to the insightful suggestions, we embarked on further analysis to ascertain the determining factors for successful lymph node dissection—whether it's the ICG imaging, the robotic method, or a synergy of both.
To elucidate this, we delved deeper into the data, examining the achievement rates of Proper Lymphadenectomy (PL) across different intervention groups. Our comparisons, outlined in Supplementary Table 1, are as follows:
Between the Laparoscopic surgery without the use of ICG group (G1) and the ICG-guided laparoscopic surgery group (G2), we observed a matching achievement rate for PL at 76.0%.
Similarly, a juxtaposition of the ICG-guided laparoscopic surgery group (G2) and the ICG-guided robotic surgery group (G3) yielded PL rates of 80.3% and 90.9%, respectively. While this denotes a notable difference, statistical significance was only realized when contrasting G1 with G3, where G3 manifested a markedly superior PL (P=0.027).
Supplementary Table 1 Proper lymphadenectomy achievement by intervention group
|
Variables, n (%) |
PL (-) |
PL (+) |
p-value |
|
Intervention group |
|||
|
G1 vs G2 |
69 (24.0%) vs 12 (19.7%) |
219 (76.0%) vs 49 (80.3%) |
0.471 |
|
G2 vs G3 |
12 (19.7%) vs 4 (9.1%) |
49 (80.3%) vs 40 (90.9%) |
0.137 |
|
G3 vs G1 |
4 (9.1%) vs 69 (24.0%) |
40 (90.9%) vs 219 (76.0%) |
0.027 |
PL, proper lymphadenectomy
Furthermore, our multivariate analysis, which aimed to discern the factors underpinning PL by juxtaposing the groups in pairs, didn't yield significant discrepancies between G1 and G2 (p=0.593) or between G2 and G3 (p=0.282). Yet, in a comparison of G1 against G3, the latter emerged as a significant factor, boasting an odds ratio of 3.156 (CI 1.076-9.256, p=0.036) (Supplementary Tables 2-1, 2-2, and 2-3).
Supplementary Table 2-1 Multivariate analysis of factors contributing to proper lymphadenectomy between intervention group (G1 vs G2)
|
Variables |
Univariate |
Multivariate |
||||
|
OR |
95% CI |
p-value |
OR |
95% CI |
p-value |
|
|
Older age (vs. < 65) |
0.772 |
0.468-1.274 |
0.312 |
|||
|
Male sex (vs. Female) |
0.966 |
0.567-1.646 |
0.898 |
|
|
|
|
ECOG 2-4 (vs. 0-1) |
0.713 |
0.243-2.086 |
0.536 |
|
|
|
|
Higher BMI (vs. <23) |
0.966 |
0.581-1.609 |
0.895 |
|
|
|
|
No comorbidity (vs.present) |
0.124 |
0.359-1.132 |
0.638 |
|
|
|
|
No history of abdominal surgery (vs. present) |
0.685 |
0.393-1.195 |
0.183 |
|
|
|
|
TG (vs. STG) |
0.566 |
0.288~1.111 |
0.098 |
|
|
|
|
T stage |
|
|
|
|
|
|
|
T1 |
Ref |
  |
Ref |
  |
||
|
T2 |
2.240 |
0.944-5.315 |
0.067 |
1.572 |
0.592-4.177 |
0.364 |
|
T3 |
2.329 |
1.125-4.822 |
0.023* |
2.491 |
1.031-6.019 |
0.043* |
|
T4 |
2.688 |
1.157-6.246 |
0.022* |
2.419 |
0.972-6.016 |
0.057 |
|
N stage |
  |
|||||
|
N0 |
Ref |
  |
Ref |
|||
|
N1 |
2.074 |
1.016-4.236 |
0.045* |
2.074 |
1.016-4.236 |
0.045* |
|
N2 |
0.914 |
0.489-1.709 |
0.779 |
0.914 |
0.489-1.709 |
0.779 |
|
N3 |
3.879 |
1.560-9.646 |
0.004* |
3.879 |
1.560-9.646 |
0.004* |
|
Intervention group |
||||||
|
G1 |
Ref |
  |
Ref |
|||
|
G2 |
1.287 |
0.647~2.557 |
0.472 |
1.214 |
0.596-2.474 |
0.593 |
OR, odds ratio; CI, confidence interval; ECOG, Eastern Cooperative Oncology Group performance status; BMI, body mass index; TG, total gastrectomy; STG, subtotal gastrectomy.
Supplementary Table 2-2 Multivariate analysis of factors contributing to proper lymphadenectomy between intervention group (G2 vs G3)
|
Variables |
Univariate |
Multivariate |
||||
|
OR |
95% CI |
p-value |
OR |
95% CI |
p-value |
|
|
Older age (vs. < 65) |
1.030 |
0.343-3.091 |
0.958 |
|||
|
Male sex (vs. Female) |
1.788 |
0.470-6.802 |
0.394 |
|
|
|
|
ECOG 2-4 (vs. 0-1) |
|
|
1.000 |
|
|
|
|
Higher BMI (vs. <23) |
1.378 |
0.455-4.174 |
0.571 |
|
|
|
|
No comorbidity (vs.present) |
0.642 |
0.215-1.917 |
0.427 |
|
|
|
|
No history of abdominal surgery (vs. present) |
1.256 |
0.325-4.847 |
0.741 |
|
|
|
|
TG (vs. STG) |
1.536 |
0.505-4.673 |
0.450 |
|
|
|
|
T stage |
|
|
|
|
|
|
|
T1 |
Ref |
  |
Ref |
  |
||
|
T2 |
|
|
0.999 |
|
|
0.999 |
|
T3 |
|
|
0.999 |
|
|
0.999 |
|
T4 |
|
|
0.999 |
|
|
0.999 |
|
N stage |
  |
|||||
|
N0 |
Ref |
  |
Ref |
|||
|
N1 |
2.250 |
0.593-8.532 |
0.233 |
3.567 |
0.752-16.914 |
0.109 |
|
N2 |
1.875 |
0.490-7.179 |
0.359 |
2.049 |
0.444-9.452 |
0.358 |
|
N3 |
|
|
0.998 |
|
|
0.998 |
|
Intervention group |
||||||
|
G2 |
Ref |
  |
Ref |
|||
|
G3 |
2.449 |
0.733-8.181 |
0.146 |
2.075 |
0.549-7.850 |
0.282 |
OR, odds ratio; CI, confidence interval; ECOG, Eastern Cooperative Oncology Group performance status; BMI, body mass index; TG, total gastrectomy; STG, subtotal gastrectomy.
Supplementary Table 2-3 Multivariate analysis of factors contributing to proper lymphadenectomy between intervention group (G1 vs G3)
|
Variables |
Univariate |
Multivariate |
||||
|
OR |
95% CI |
p-value |
OR |
95% CI |
p-value |
|
|
Older age (vs. < 65) |
0.723 |
0.429-1.219 |
0.224 |
|||
|
Male sex (vs. Female) |
0.922 |
0.532-1.596 |
0.771 |
|
|
|
|
ECOG 2-4 (vs. 0-1) |
0.603 |
0.203-1.795 |
0.364 |
|
|
|
|
Higher BMI (vs. <23) |
1.058 |
0.622-1.798 |
0.836 |
|
|
|
|
No comorbidity (vs.present) |
0.590 |
0.324-1.075 |
0.085 |
|
|
|
|
No history of abdominal surgery (vs. present) |
0.779 |
0.436-1.394 |
0.401 |
|
|
|
|
TG (vs. STG) |
0.526 |
0.245-1.089 |
0.084 |
|
|
|
|
T stage |
|
|
|
|
|
|
|
T1 |
Ref |
  |
Ref |
  |
||
|
T2 |
1.826 |
0.771-4.324 |
0.171 |
1.408 |
0.525-3.779 |
0.496 |
|
T3 |
1.984 |
0.968-4.067 |
0.061 |
2.344 |
0.941-5.840 |
0.067 |
|
T4 |
2.538 |
1.068-6.030 |
0.035* |
2.740 |
1.053-7.128 |
0.039* |
|
N stage |
  |
|||||
|
N0 |
Ref |
  |
Ref |
|||
|
N1 |
1.990 |
0.922-4.297 |
0.080 |
2.001 |
0.922-4.342 |
0.079 |
|
N2 |
0.972 |
0.509-1.855 |
0.931 |
0.949 |
0.464-1.824 |
0.876 |
|
N3 |
4.048 |
1.619-10.124 |
0.003* |
3.965 |
1.578-9.961 |
0.003* |
|
Intervention group |
||||||
|
G1 |
Ref |
  |
Ref |
|||
|
G3 |
3.151 |
1.088-9.120 |
0.034* |
3.156 |
1.076-9.256 |
0.036* |
OR, odds ratio; CI, confidence interval; ECOG, Eastern Cooperative Oncology Group performance status; BMI, body mass index; TG, total gastrectomy; STG, subtotal gastrectomy
This extended analysis illuminates that while neither ICG imaging nor the robotic method individually exerted a pronounced impact on PL, their combined application in the ICG-guided robotic surgery group showcased significant efficacy in achieving PL, particularly when juxtaposed against the Laparoscopic surgery without the use of ICG group. Thus, reinforcing our contention that ICG-robotic gastrectomy augments lymphadenectomy outcomes in select gastric cancer instances, heralding it as a propitious surgical avenue for forthcoming endeavors.
We're indebted to the reviewer for prompting this supplementary analysis, which has undeniably enriched the depth and dimension of our study.
We have added the following section to the result (page 5.Line 142-152).
[In this study, proper lymphadenectomy (PL) was characterized as the retrieval of more than 30 ELNs. Out of the entire cohort, 308 patients (78.4%) achieved PL, while the remaining 85 (21.6%) did not meet this threshold. Impressively, the G3 group, which un-derwent ICG-guided robotic surgery, demonstrated a PL rate of 90.9%, thus surpassing the other groups. Subsequent analyses showed that patients with a higher N stage and more advanced cancer stages were more likely to achieve over 30 ELNs (Table 2). A closer as-sessment between the groups revealed a PL achievement rate of 76.0% for G1, compared to 80.3% for G2. The comparison between G2 and G3 yielded PL rates of 80.3% and 90.9%, respectively. While these rates differ, the difference was not statistically significant. How-ever, when comparing G1 with G3, a significant difference in PL rates emerged, favoring G3 (P=0.027), as detailed in Supplementary Table 1.]
We have added the following section to the result (page 6.Line 160-165).
[Further scrutiny of our multivariate analysis, aimed at identifying specific factors associ-ated with PL, revealed no significant differences between the G1 and G2 groups (p=0.593) or between the G2 and G3 groups (p=0.282). Interestingly, a comparison of G1 against G3 highlighted the G3 group as a significant predictor for achieving PL, demonstrating an odds ratio of 3.156 (CI 1.076-9.256, p=0.036) (Supplementary Tables 2-1, 2-2, and 2-3).]
We also have added the following section to the discussion (page 9.Line 241-251).
[Diving deeper into the factors influencing proper lymphadenectomy, our extended analy-sis did not reveal statistically significant differences between groups G1 and G2 or G2 and G3. However, a notable difference was observed when comparing G3 to G1. This suggests that while the individual impact of ICG imaging or robotic surgery may not be readily discernible, their combined utilization in the ICG-guided robotic surgery group signifi-cantly elevates the success rate of proper lymphadenectomy. Specifically, when juxta-posed against the Laparoscopic surgery without the use of ICG group, the synergy of these methods becomes evident. This lends further credence to our stance that ICG-robotic gas-trectomy can potentially enhance lymphadenectomy outcomes in selected gastric cancer cases, marking it as a promising surgical approach for forthcoming applications.]
Round 2
Reviewer 2 Report
Authors revised the article according to reviewer’s comments appropriately.